

# Pleistocene phylogeography and cryptic diversity of a tiger beetle, *Calomera littoralis*, in North-Eastern Mediterranean and Pontic regions inferred from mitochondrial COI gene sequences

Radomir Jaskuła[1], Tomasz Rewicz[2], Mateusz Płóciennik[1] and Michał Grabowski[1]

[1] Department of Invertebrate Zoology and Hydrobiology, University of Lodz, Łódź, Poland
[2] Laboratory of Microscopic Imaging and Specialized Biological Techniques, University of Lodz, Łódź, Poland

## ABSTRACT

**Background.** *Calomera littoralis* is a Palearctic species, widely distributed in Europe; inhabiting predominantly its Atlantic, Mediterranean and Black Sea coastlines.
**Methods.** Its phylogeography on the Balkan Peninsula and on the north-western Black Sea coast was inferred using a 697 bp long portion of the mitochondrial COI gene, amplified from 169 individuals collected on 43 localities.
**Results.** The results revealed two genetically divergent groups/lineages, the southern one inhabiting both the Balkan Peninsula and the Pontic Region and the northern one found exclusively in the Pontic Region. Species delimitation based on DNA barcoding gap suggested an interspecific level of divergence between these groups. Multivariate analysis of eight male and female morphometric traits detected no difference between the groups, implying they may represent cryptic species. The Bayesian time-calibrated reconstruction of phylogeny suggested that the lineages diverged ca. 2.3 Ma, in early Pleistocene.
**Discussion.** The presence of the two genetically divergent groups results most likely from contemporary isolation of the Pontic basin from the Mediterranean that broke the continuous strip of coastal habitats inhabited by *C. littoralis*. Demographic analyses indicated that both lineages have been in demographic and spatial expansion since ca. 0.15 Ma. It coincides with the terminal stage of MIS-6, i.e., Wartanian/Saalian glaciation, and beginning of MIS-5e, i.e., Eemian interglacial, during which, due to eustatic sea level rise, a wide connection between Mediterranean and the Pontic basin was re-established. This, along with re-appearance of coastal habitats could initiate north-east expansion of the southern lineage and its secondary contact with the northern one. The isolation of the Pontic basin from the Mediterranean during the Weichselian glaciation most likely did not have any effect on their phylogeography.

Corresponding author
Radomir Jaskuła,
radekj@biol.uni.lodz.pl

## INTRODUCTION

The Eastern Mediterranean, including the Pontic area, is recognised as one of the major biodiversity and endemism hot spots on a global scale, as well as a major glacial refugium in Europe (e.g., *Myers et al.*, *2000*; *Kotlík, Bogutskaya & Ekmekçi*, *2004*; *Blondel et al.*, *2010*). Among others, it is a consequence of complex geological history of the region that was an archipelago and united with rest of the European continent only in Neogene (*Pfiffner*, *2014*). On the other hand, a shallow epicontinental sea, Paratethys, occupied vast areas of the continent and regressed gradually leaving relics, such as Black, Azov and Caspian Sea (*Nahavandi et al.*, *2013*). Local isostatic and eustatic changes of sea level were among superior phenomena shaping local landscapes. For example, there were at least twelve saline water intrusions from the Mediterranean Sea, and eight intrusions from the Caspian Lake to the Black Sea during the last 0.67 million years (Ma) i.e., in Pleistocene (*Badertscher et al.*, *2011*). Inevitably, they played an important role in modelling diversity and distribution patterns for numerous organisms, particularly those inhabiting coastal ecosystems both in the Mediterranean and in the Pontic area. However, the evidence comes mostly from aquatic, predominantly marine or brackish water, taxa (e.g., *Audzijonyte, Daneliya & Vainola*, *2006*; *Neilson & Stepien*, *2011*). There is a deficiency of studies focusing upon coastal species inhabiting terrestrial habitats in this region (*Akin et al.*, *2010*).

Tiger beetles, Cicindelidae Latreille, 1806, seem to be ideal model organisms to test such assumptions. The family, with more than 2,600 species, has a worldwide distribution with exception of polar regions and some oceanic islands (*Pearson & Cassola*, *2005*). Most species, both in larval and adult stage, prefer various types of sandy areas and are habitat specialists; often inhabiting coastal areas (*Pearson & Vogler*, *2001*). Several studies dealt with phylogeography of tiger beetles in various regions of the world (e.g., *Vogler et al.*, *1993*; *Cardoso & Vogler*, *2005*; *Woodcock et al.*, *2007*), yet so far only few focused on the role of sea level oscillations in their evolutionary history (*Vogler & DeSalle*, *1993*; *Sota et al.*, *2011*) or compared the diversity patterns on both, the molecular and morphological, levels (*Cardoso, Serrano & Vogler*, *2009*; *Tsuji et al.*, *2016*).

The tiger beetle, *Calomera littoralis* (*Fabricius*, *1787*), is widely distributed in Palaearctic, from the Iberian Peninsula and Morocco in the west to the Middle Asia and Russian Far East in the east (*Putchkov & Matalin*, *2003*; *Serrano*, *2013*; *Jaskuła*, *2011*; *Jaskuła*, *2015*). Generally, it is recognised as euryoecious (*Jaskuła*, *2011*; *Jaskuła*, *2013*; *Jaskuła*, *2015*). However, in Europe it occupies predominantly the very narrow stretch of Atlantic, Mediterranean and Black Sea coastal habitats (*Cassola & Jaskuła*, *2004*; *Franzen*, *2006*; *Jaskuła*, *2007a*; *Jaskuła*, *2007b*; *Jaskuła, Pešić & Pavicević*, *2005*; *Serrano*, *2013*).

Taking into account the history of recurrent closing and reopening of the connection between the Mediterranean and the Black Sea in the Pleistocene, we hypothesised that it should leave a signature in genetic and possibly morphological polymorphism of *Calomera littoralis*, which is commonly found around both sea basins. Thus, we aimed at (1) exploring and comparing spatial patterns of molecular and morphological diversity of this species in the Mediterranean and Pontic region, (2) interpreting the observed patterns in the context of local paleogeography.

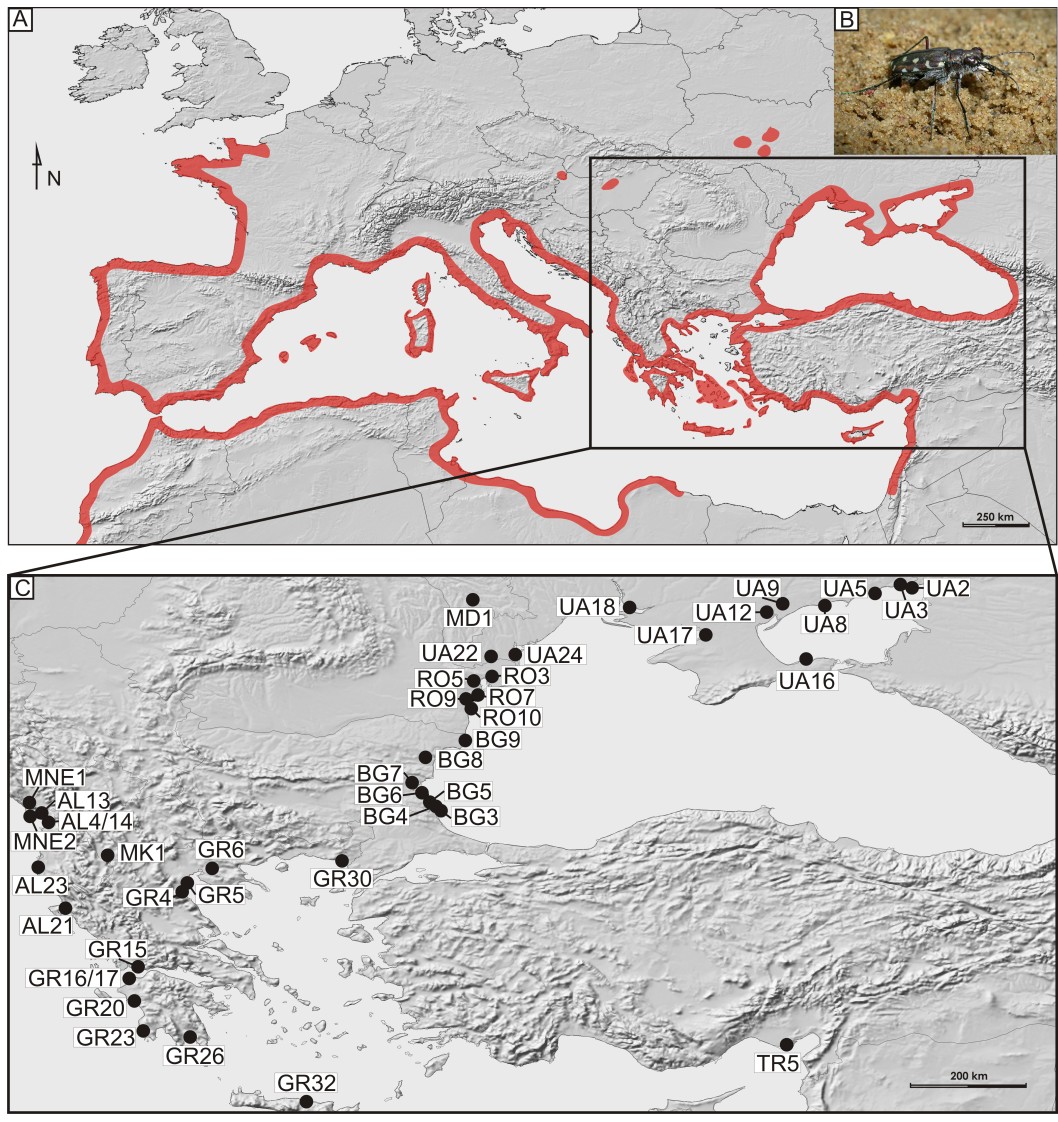

**Figure 1** **Distribution and sampling of *Calomera littoralis* in Europe.** (A) General distribution of *Calomera littoralis* in Europe shown as red-shaded area. (B) Picture of *Calomera littoralis* beetle. (C) Sampling sites in Balkan Peninsula, Black Sea region and Turkey shown as black dots. Localities coded as in Table 1.

# MATERIAL AND METHODS

## Sample collection and identification

In total, 169 imagines of *Calomera littoralis* were collected with entomological hand net on 43 sites on the Mediterranean coasts of the Balkan Peninsula, Crete and Turkey as well as on the northern and western coast of the Black and Azov Seas, in the years 2009–2012 (Fig. 1 and Table 1). At a site the material was fixed in 96% ethanol for DNA preservation. Taxonomic identification of the collected material followed *Mandl* (*1981*)

Peer J

**Table 1** Sampling localities for *Calomera littoralis* in the North-Eastern Mediterranean and Pontic regions.

| Abbr. | Locality | Country | Coordinates | | OTU ABGD | N | COI haplotypes | Acc. nos. COI |
|-------|----------|---------|-------------|----------|----------|---|----------------|----------------|
| | | | Longitude | Latitude | | | | |
| AL04 | Lezhe | Albania | 19.60032 | 41.77051 | SL | 2 | H1(1), H2(1) | KU905303–KU905304 |
| AL13 | Velipoja | Albania | 19.44742 | 41.86185 | SL | 1 | H3(1) | KU905302 |
| AL14 | Lezhe | Albania | 19.60026 | 41.77029 | SL | 3 | H1(1), H4(2) | KU905299–KU905301 |
| AL23 | Butrint | Albania | 20.00576 | 39.74292 | SL | 5 | H5(2), H6(1), H7(1), H8(1) | KU905294–KU905298 |
| BG03 | Sinemorec | Bulgaria | 27.97311 | 42.06318 | NL | 4 | H9(1), H10(1), H11(1), H12(1) | KU905217–KU905220 |
| BG04 | Achtopol | Bulgaria | 27.92366 | 42.10304 | SL, NL | 3 | H13(1), H14(1), H15(1) | KU905214–KU905216 |
| BG05 | Carewo | Bulgaria | 27.87794 | 42.14655 | SL | 1 | H16(1) | KU905213 |
| BG06 | Dyuni | Bulgaria | 27.72104 | 42.34988 | SL, NL | 3 | H15(1), H17(1), H18(1) | KU905210–KU905212 |
| BG07 | Burgas | Bulgaria | 27.48438 | 42.55187 | SL, NL | 6 | H11(1), H14(2), H16(1), H19(1), H20(1) | KU905204–KU905209 |
| BG08 | Beloslav | Bulgaria | 27.73240 | 43.19124 | SL, NL | 6 | H13(1), H18(1), H21(1), H22(1), H23(1), H24(1) | KU905198–KU905203 |
| BG09 | Shabla | Bulgaria | 28.58338 | 43.57218 | SL, NL | 6 | H11(1), H15(1), H16(1), H25(1), H(26), H(27) | KU905192–KU905197 |
| GR04 | Limani Litochorou | Greece | 22.54858 | 40.15725 | SL | 2 | H28(1), H29(1) | KU905292–KU905293 |
| GR05 | Katerini | Greece | 22.61182 | 40.29430 | SL | 5 | H7(1), H30(2), H31(1), H32(1) | KU905287–KU905291 |
| GR06 | Agios Vasileios | Greece | 23.16222 | 40.65620 | SL | 1 | H29(1) | KU905286 |
| GR15 | Kokori | Greece | 21.55359 | 38.37430 | SL | 6 | H7(1), H29(3), H30(1), H33(1) | KU905280–KU905285 |
| GR16 | Akrotiri Araksou | Greece | 21.39320 | 38.18333 | SL | 2 | H34(1), H35(1) | KU905278–KU905279 |
| GR17 | Kalogria | Greece | 21.38517 | 38.15959 | SL | 5 | H7(1), H30(1), H34(1), H36(1), H37(1) | KU905273–KU905277 |
| GR20 | Pyrgos | Greece | 21.47691 | 37.64011 | SL | 6 | H29(1), H30(2), H34(2), H38(1) | KU905267–KU905272 |
| GR23 | Gialova | Greece | 21.69121 | 36.95367 | SL | 2 | H1(2) | KU905265–KU905266 |
| GR26 | Evrotas river mouth | Greece | 22.69421 | 36.80451 | SL | 4 | H29(2), H30(2) | KU905256–KU905259 |
| GR30 | Evros river mouth | Greece | 25.97922 | 40.82814 | SL | 5 | H7(1), H30(1), H39(1), H40(1), H41(1) | KU905260–KU905264 |
| GR32 | Karteros | Greece | 25.19224 | 35.33255 | SL | 6 | H29(4), H42(2) | KU905250–KU905255 |
| MD01 | Molesti | Moldova | 28.754521 | 46.789716 | SL, NL | 6 | H11(1), H24(1), H26(1), H43(1), H44(1), H45(1) | KU905179–KU905184 |
| MK01 | Stenje | Macedonia | 20.90385 | 40.94522 | SL | 4 | H7(2), H29(1), H46(1) | KU905175–KU905178 |
| MNE01 | Donji Murići | Montenegro | 19.22248 | 42.16319 | SL | 4 | H4(1), H47(1), H48(2) | KU905188–KU905191 |
| MNE02 | Doni Štoj | Montenegro | 19.33309 | 41.87111 | SL | 3 | H29(2), H48(1) | KU905185–KU905187 |

Jaskuła et al. (2016), *PeerJ*, DOI 10.7717/peerj.2128

**Table 1** (*continued*)

| Abbr. | Locality | Country | Coordinates | | OTU ABGD | N | *COI* haplotypes | Acc. nos. *COI* |
|---|---|---|---|---|---|---|---|---|
| | | | Longitude | Latitude | | | | |
| RO03 | Murihiol | Romania | 29.16071 | 45.02292 | NL | 6 | H12(1), H19(1), H49(1), H50(1), H51(2), H52(1) | KU905244–KU905249 |
| RO05 | Enisala | Romania | 28.80822 | 44.88047 | SL, NL | 6 | H10(1), H14(2), H17(1), H18(1), H53(1) | KU905238–KU905243 |
| RO07 | Sinoe | Romania | 28.79436 | 44.62350 | SL, NL | 5 | H14(1), H22(1), H54(1), H55(1), H56(1) | KU905233–KU905237 |
| RO09 | Istria | Romania | 28.72625 | 44.53820 | NL | 6 | H11(2), H14(1), H22(1), H57(1), H58(1) | KU905227–KU905232 |
| RO10 | Corbu | Romania | 28.71192 | 44.37732 | NL | 6 | H11(1), H53(1), H59(1), H60(1), H61(1), H62(1) | KU905221–KU905226 |
| TR05 | Bebeli | Turkey | 35.47895 | 36.62488 | SL | 4 | H7(1), H63(2), H64(1) | KU905171–KU905174 |
| UA02 | Siedowe | Ukraine | 38.12819 | 47.07738 | NL | 4 | H15(1), H51(1), H65(1), H66(1) | KU905336–KU905339 |
| UA03 | Samsonowe | Ukraine | 38.01095 | 47.09550 | SL, NL | 2 | H67(1), H68(1) | KU905334– KU905335 |
| UA05 | Melekyne | Ukraine | 37.38399 | 46.94367 | SL, NL | 2 | H69(1), H70(1) | KU905332– KU905333 |
| UA08 | Preslav | Ukraine | 36.29574 | 46.66028 | NL | 1 | H25(1) | KU905331 |
| UA09 | Hirsivka | Ukraine | 35.34955 | 46.65631 | NL | 12 | H11(2), H12(2), H18(1), H26(1), H71(1), H72(2), H73(2), H74(1) | KU905319– KU905330 |
| UA12 | Davydivka | Ukraine | 35.11976 | 46.50789 | SL, NL | 5 | H7(1), H75(1), H76(1), H77(1), H78(1) | KU905314–KU905318 |
| UA16 | Azovsk | Ukraine | 35.88406 | 45.40428 | NL | 1 | H11(1) | KU905313 |
| UA17 | Tavirsk | Ukraine | 33.72799 | 45.97222 | NL | 3 | H18(1), H50(1), H79(1) | KU905310–KU905312 |
| UA18 | Oleksandrivka | Ukraine | 32.11789 | 46.60185 | NL | 3 | H18(1), H80(1), H81(1) | KU905307–KU905309 |
| UA22 | Komyshivka | Ukraine | 29.14931 | 45.48260 | NL | 1 | H10(1) | KU905306 |
| UA24 | Prymorsk | Ukraine | 29.65798 | 45.53664 | SL | 1 | H17(1) | KU905305 |

### DNA extraction, amplification and sequencing

Following *Hillis, Moritz & Mable* (*1996*) the standard phenol–chloroform method was used to extract DNA from all the collected individuals. Air-dried DNA pellets were eluted in 100 μl of TE buffer, pH 8.00, stored at 4 °C until amplification, and subsequently at −20 °C for long-term storage.

Fragments of mitochondrial cytochrome oxydase subunit I gene (COI), ca. 700 bp long, were amplified using the Jerry and Pat pair of primers (*Simon et al.*, *1994*). Each PCR reaction was conducted in a total volume of 10 μl and contained DreamTaq Master Mix (1x) Polymerase (ThermoScientific), 200 nM of each primer and 1 μl of DNA template. The thermal regime consisted of initial denaturation at 94 °C for 2 min, followed by 34 cycles of denaturation at 94 °C for 30 s, annealing at 44 °C for 30 s, and elongation at 72 °C for 60 s, completed by a final extension at 72 °C for 10 min. The amplified products were visualized on 2.0% agarose gels stained with MidoriGreen (Nippon Genetics) to verify the quality of the PCR reactions. Then, the PCR products were chemically cleaned up of dNTPs and primer residues by adding 5U of Exonuclease I (Thermo Scientific) and 1U of FastAP Alkaline Phosphatase (Thermo Scientific) per sample. The COI amplicon was sequenced one way using BigDye sequencing protocol (Applied Biosystems 3730xl) by Macrogen Inc., Korea.

### Molecular data analysis

First, all the obtained sequences were positively verified as *Calomera* DNA using GenBankBLASTn searches (*Altschul et al.*, *1990*). They were then edited and assembled with CLUSTALW algorithm (*Chenna et al.*, *2003*) using BIOEDIT© 7.2.5. The resulting alignment was 697 bp long with no gaps, and composed of 169 COI sequences. The sequence data and trace files were uploaded to BOLD and subsequently also to GenBank (accession numbers KU905171–KU905339).

Pairwise Kimura 2-parameter (K2p) distances between sequences were estimated using MEGA 6.2 (*Tamura et al.*, *2013*). Haplotypes were retrieved using DNASP v5 (*Librado & Rozas*, *2009*). Phylogenetic relationships between the haplotypes were visualised with phylogenetic network computed using the neighbour-net algorithm and uncorrected p-distances in SplitsTree ver. 4.13.1 (*Huson & Bryant*, *2006*).

To test for presence of distinct operational taxonomic units (OTUs) that may represent potential cryptic species/subspecies in the sequenced pool of individuals we used the Automatic Barcode Gap Discovery (ABGD) procedure (*Puillandre et al.*, *2012*). The default value of 0.001 was used as the minimum allowed intraspecific distance. The maximum allowed intraspecific distance was set to $P\text{max} = 0.03$ and 0.06, as both threshold values have been already used in literature to delimit insect species (*Hebert et al.*, *2003*; *Hebert, Ratnasingham & DeWaard*, *2003*). We applied the K2p model sequence correction, which is a standard for barcode analyses (*Hebert et al.*, *2003*). We used primary partitions as a principal for group definition for they are usually stable over a wider range of prior values, minimise the number of false positive (over split species) and are usually close to the number of groups described by taxonomists (*Puillandre et al.*, *2012*).

To reveal the temporal framework for the divergence of the OTUs (potential cryptic species) defined within *Calomera littoralis*, the time calibrated phylogeny was reconstructed in BEAST, version 1.8.1 (*Drummond et al.*, *2012*). A COI sequence of *Calomera lugens aphrodisia* Baudi di Selve 1864 from GenBank (accession number KC963733) was used as an outgroup. This analysis was performed on a reduced dataset, containing only the most distant haplotypes from each OTU. Hasegawa–Kishino–Yano (HKY) model of evolution, selected as best-fitting to our dataset in MEGA 6.2, and coalescent model were set as tree priors. The strict clock with rate 0.0115, widely used for phylogenetic studies upon insects, was applied for the analyses (*Brower*, *1994*). Five runs of 20 M iterations of Markov chain Monte Carlo (MCMC) sampled each 2000 iterations were performed. The runs were examined using Tracer v 1.6 and all sampled parameters achieve sufficient effective sample sizes (ESS > 200). Tree files were combined using Log-Combiner 1.8.1 (*Drummond et al.*, *2012*), with removal of the non-stationary 20% burn-in phase. The maximum clade credibility tree was generated using TreeAnnotator 1.8.1 (*Drummond et al.*, *2012*).

To provide insight into historical demography, i.e., the temporal changes of the effective population size of *Calomera littoralis* in the studied region, we performed Bayesian Skyline Plot (BSP) analysis (*Drummond et al.*, *2005*) in BEAST, version 1.8.1 (*Drummond et al.*, *2012*). Separate analysis was performed for each of the two phylogenetic lineages revealed in our study (see 'Results'). The Northern Lineage was represented by 84 individuals from 22 localities, while the Southern Lineage was represented by 85 individuals from 32 localities. The HKY+I model of evolution was used as the best fitting model in case of the Eastern Lineage, while TN93+I was used in case of the Western Lineage. Two runs of MCMC, 20 M iterations long sampled each 2000 iterations, were performed. In both cases the runs were examined using Tracer v 1.6 (*Drummond et al.*, *2012*) and all sampled parameters achieved sufficient effective sample sizes (ESS > 200).

Two models of population expansion, demographic and spatial, were examined using mismatch distribution analysis (*Slatkin & Hudson*, *1991*; *Rogers & Harpending*, *1992*) and Tajima's $D$ neutrality test (*Tajima*, *1989*). Analyses were performed for the COI groups, using Arlequin 3.5.1.3 (*Excoffier & Lischer*, *2010*) with 1,000 replicates.

## Morphometric data analysis

To test whether variation of morphometric traits reflects presence of two genetically divergent lineages (potential cryptic species), measurements of eight body parameters (Fig. 2) were taken from all the 69 males and 100 females used previously for the molecular analyses: 1, right mandible length (RML); 2, length of head (LH); 3, width of head (WH); 4, pronotum length (PL); 5, maximum pronotum width (MPW); 6, elytra length (EL); 7, maximum elytra width (MEW); and 8, total body length (TBL). The Principal Component Analysis (PCA) was performed separately for each sex (Fig. 3). To test for significance ($p < 0.01$) of morphological differences (separately for males and females) between the two divergent lineages one-way ANOSIM Pairwise Test was performed. All the above statistical analyses were done with PRIMER 6 software (*Clarke & Gorley*, *2006*).

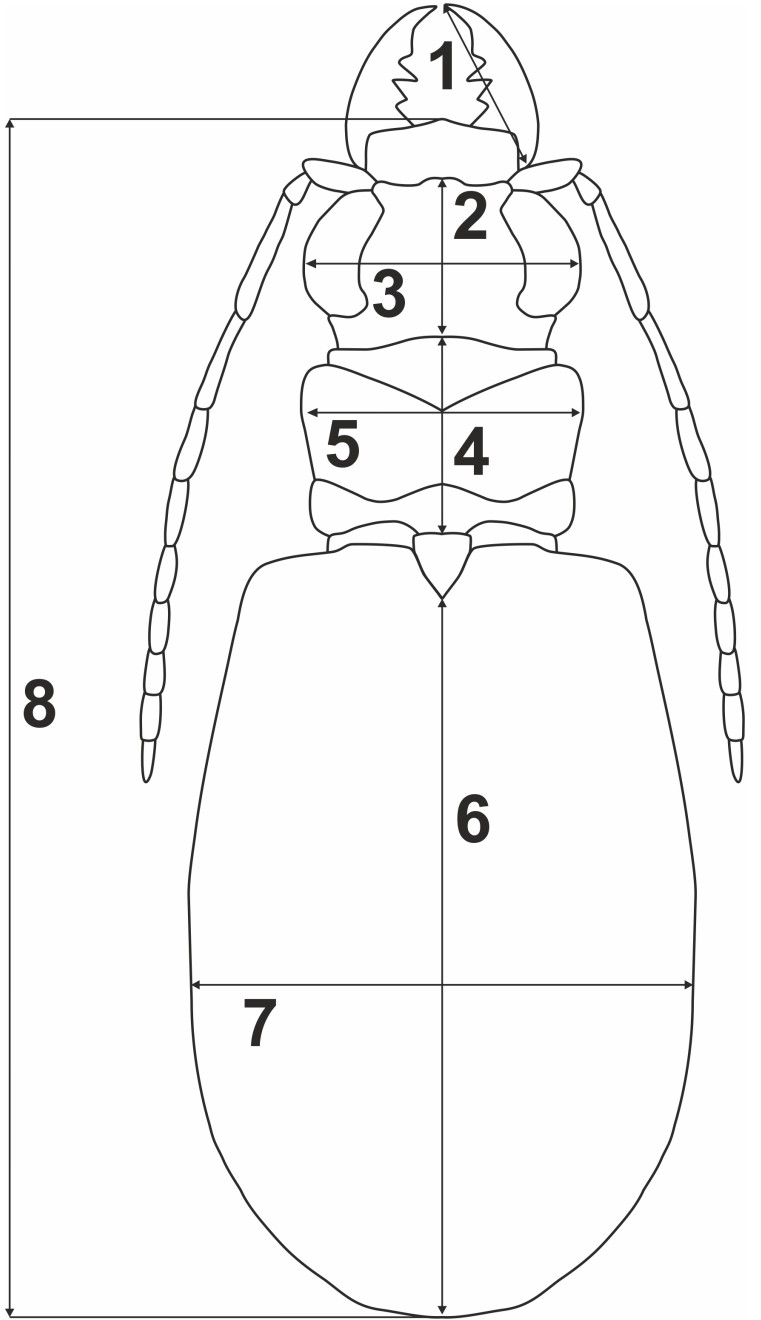

**Figure 2  Body parameters measured in *Calomera littoralis*.** 1, RML—right mandible length; 2, LH—length of head; 3, WH—width of head; 4, PL—pronotum length; 5, MPW—maximum pronotum width; 6, EL—elytra length; 7, MEW—maximum elytra width; 8, TBL—total body length.

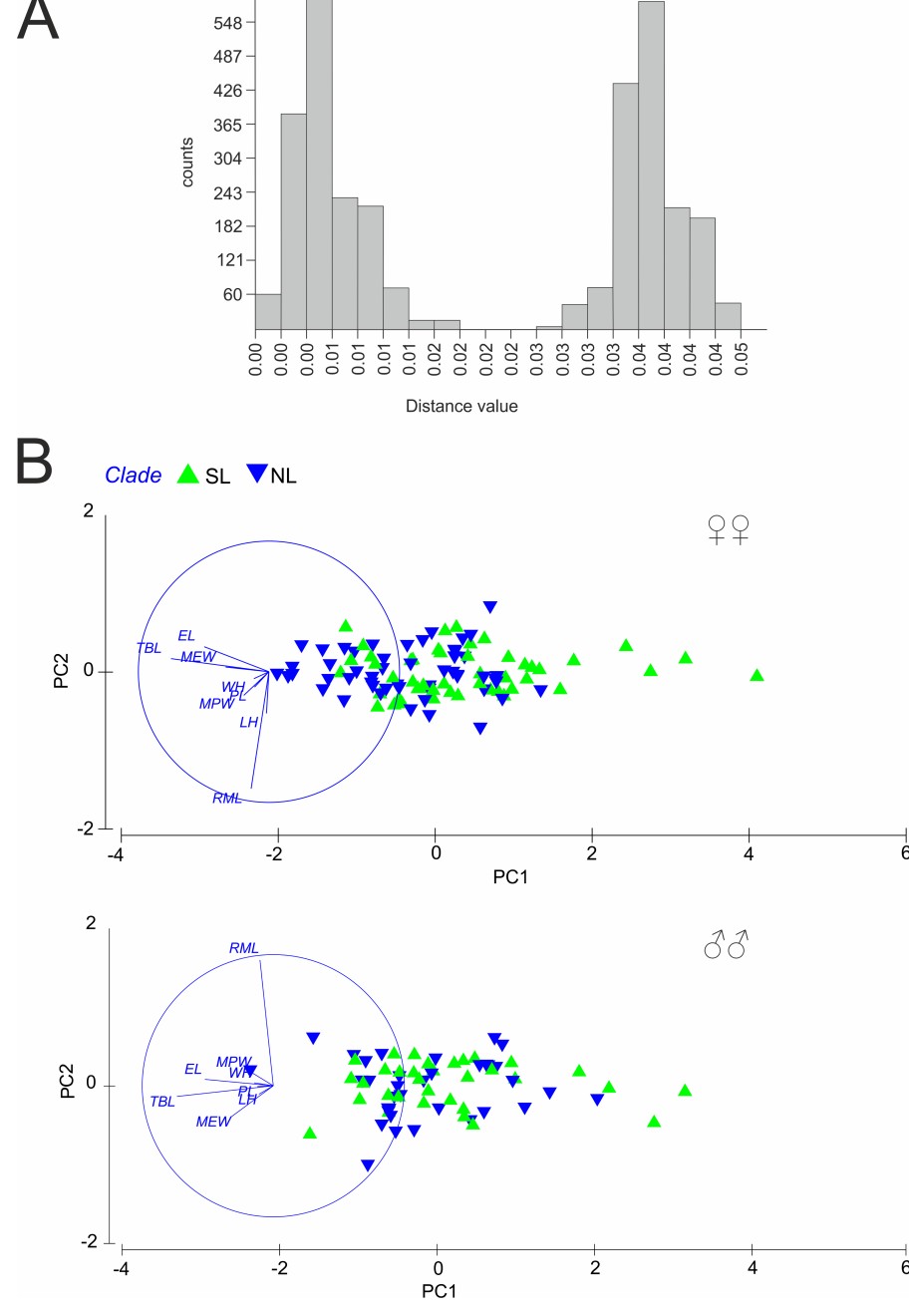

**Figure 3** (A) **Automatic Barcode Gap Discovery (ABGD) analysis of *Calomera littoralis* and (B) Results of Principal Component Analysis performed for investigated specimens on main body dimensions.** SL, southern lineage; NL, northern lineage; RML, right mandible length; WH, width of head; LH, length of head; MPW, maximum pronotum width; PL, pronotum length; EL, elytra length; MEW, maximum elytra width; TBL, total body length. Both in ABGD and PCA analyses 169 specimens from 43 sites from the Mediterranean and the Pontic areas were used.

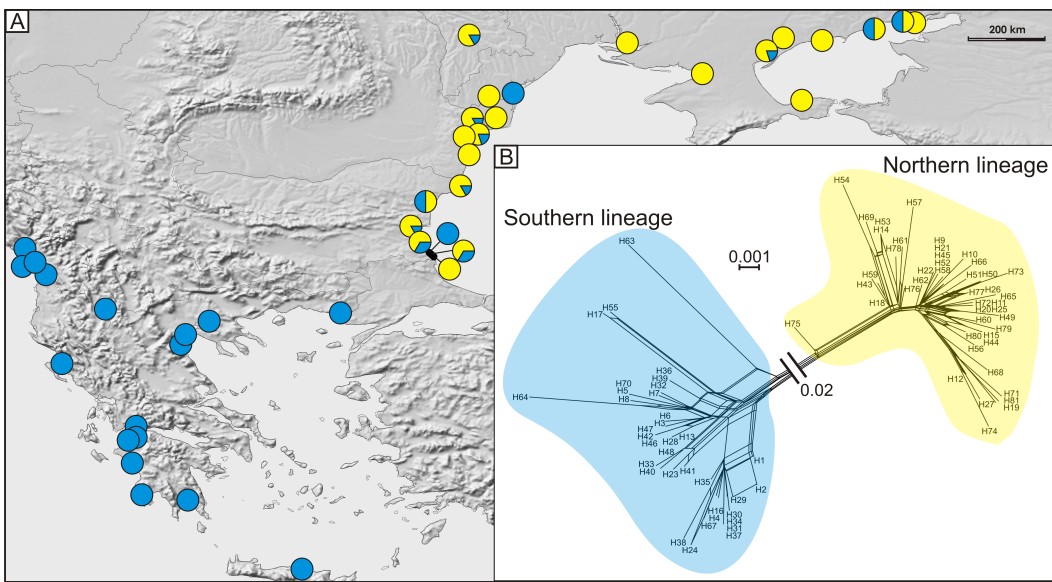

**Figure 4** (A) Geographic distribution of haplogroups from southern (blue circles) and northern (yellow circles) lineages and (B) median joining network of 81 detected COI haplotypes showing southern (blue shading), and northern (yellow shading) lineages.

## RESULTS

### Molecular data

A total of 81 haplotypes were identified in the dataset composed of 169 individuals from 43 sites from the Mediterranean and the Pontic areas (Table 1). The phylogenetic network illustrating phylogenetic relationships among haplotypes (Fig. 4) uncovered presence of two distinct haplotype groups (phylogenetic lineages). The first group, from now on defined as southern lineage, includes 36 haplotypes present all over the studied range including the Balkan Peninsula and the Pontic area. The other group, from now on defined as northern lineage, is composed of 45 haplotypes present exclusively along the north-western coast of the Black Sea. The mean K2p genetic distance between both groups of haplotypes is relatively high (0.039, SD 0.007). Both variants of the ABGD analysis resulted in partitioning of the dataset into two OTUs, that may represent distinct operational taxonomic units—potential cryptic species or subspecies within *Calomera littoralis* in the studied area (Fig. 3A).

The Bayesian time-calibrated reconstruction of phylogeny shows that the two lineages split at ca. 2 Ma, i.e., in early Pleistocene (Fig. 5A). Results of the BSP analyses showing the temporal changes of the effective population size suggests that both lineages experienced rapid population growth that has started ca. 0.15 Ma (Fig. 5B). In both cases, a small decline in effective population size may be observed in most recent times (<0.05 Ma). Results of the mismatch analysis show that both lineages are currently in the stage of both demographic and spatial expansion (Fig. 5C). Interestingly, geographical distribution of both lineages shows that the spatial expansion of southern lineage was efficient enough to spread eastwards into the Black Sea and colonise effectively the north-western Black Sea coast. The northern lineage has spread only in the Pontic region.

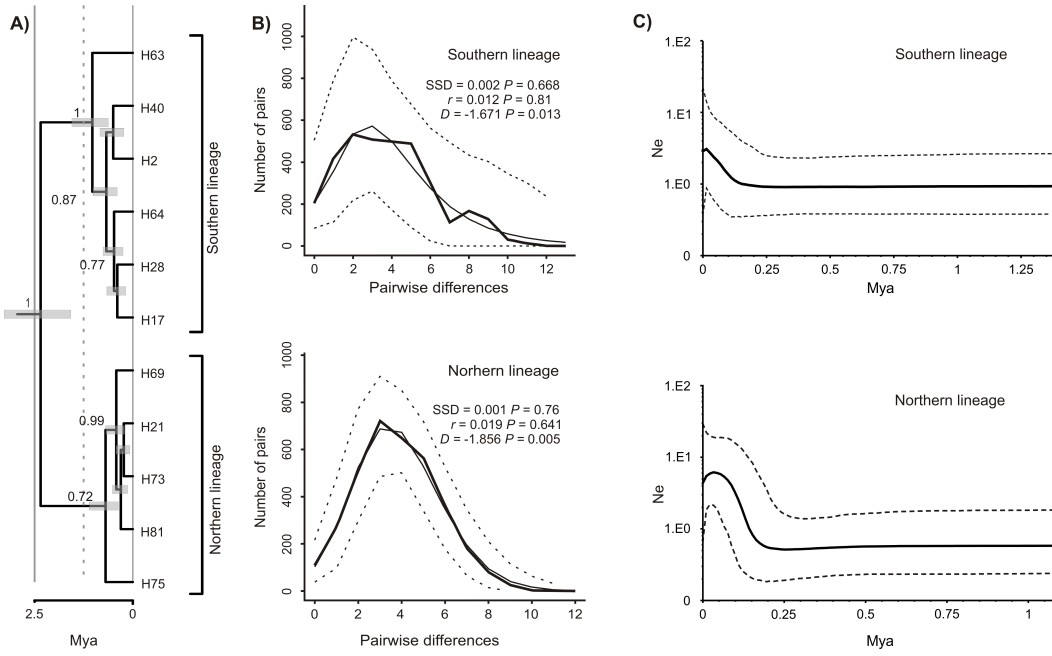

**Figure 5** **Phylogeny and historical demography of *Calomera littoralis*.** (A) Maximum clade credibility chronogram with a strict molecular clock model inferred from COI sequences. The numbers next to the respective node indicate Bayesian posterior probabilities higher than 0.5. (B) Mismatch plots for southern and northern lineage. Thin solid lines indicate expected frequency under model of population demographic expansion, thick solid lines represent observed frequency, and dashed lines indicate 95% confidence intervals for the observed mismatch. SSD, sum of squared deviation; $r$, Harpending's raggedness index; $D$, Tajima's $D$. (C) Bayesian skyline plots for southern and northern lineages of *Calomera littoralis*. Solid lines indicate the median posterior effective population size through time; dashed lines indicate the 95% highest posterior density interval for each estimate.

## Morphometric data

The results of PCA and ANOSIM revealed no differences in the analysed morphometric traits between the southern and the northern lineages, neither in males nor in females (Fig. 3B). In PCA (Fig. 3B), a very weak gradient ($R = 0.03$) could be seen in case of female body length. Females from the northern lineage clade were slightly larger than those from the southern one (body length; ANOSIM Pairwise Tests $p = 0.03$).

## DISCUSSION AND CONCLUSIONS

### Cryptic diversity of *Calomera littoralis*

Known as very important hotspot of biodiversity, endemicity and cryptic diversity (e.g., *Myers et al.*, *2000*; *Kryštufek & Reed*, *2004*; *Blondel et al.*, *2010*; *Huemer & Timossi*, *2014*; *Previšić et al.*, *2014*; *Caković et al.*, *2015*), the southern Europe holds also the most diverse tiger beetle fauna in the entire Palearctic realm (*Jaskuła*, *2011*). Presence of cryptic diversity was already pointed out for *Cicindela hybrida* in the Mediterranean (*Cardoso, Serrano & Vogler*, *2009*) as well as for several species of tiger beetles occurring in other parts of the world (*Vogler & Pearson*, *1996*; *López-López, Hudson & Galián*, *2012*). Thus, existence of well-defined OTUs within *Calomera littoralis* is not surprising in the studied area. The level

of divergence, 0.04 K2p distance, between the northern and the southern lineage is similar as those found between species of tiger beetles in other studies (e.g., *Cardoso & Vogler*, *2005*; *López-López, Abdul Aziz & Galián*, *2015*). Interestingly, we could not detect any conclusive morphological differences between the two lineages based on the multivariate analysis of eight morphometric traits. It must be mentioned that three subspecies of *Calomera littoralis*, described on the basis of morphology, were reported from the studied area: *C. l. nemoralis* from all the studied Balkan countries, Crete, Moldova, western Ukraine and western Turkey; *C. l. conjunctaepustulata* (*Doktouroff*, *1887*) from the Azov Sea area; *C. l. winkleri* (*Mandl*, *1934*) from Crete and the coastal zone of southern Turkey (*Werner*, *1991*; *Putchkov & Matalin*, *2003*; *Avgin & Özdikmen*, *2007*). However, the morphological differences between the subspecies, such as body size, maculation of elytra and shape of aedeagus, are poorly defined and did not allow the identification of the studies material further than to the species level. Unfortunately, we had no opportunity to study the topotypical material—Provence, France, is *locus typicus* for *C. l. nemoralis*, Tibet for *C. l. conjunctaepustulata*, and Cyprus for *C. l. winkleri*. Thus, we cannot exclude a possibility that the two lineages we found in our material overlap with any of the above mentioned subspecies. However, only a further taxonomic revision combining more phenotypic traits, including e.g., cuticle ultrastructure, with several, mitochondrial and nuclear DNA data, could help to resolve this problem. Until such revision is done, we propose to use the tentative name "*Calomera littoralis* complex" for populations from the studied area.

## Phylogeography of *Calomera littoralis*

Occurrence of *C. littoralis* in Europe is restricted predominantly to marine shorelines with sandy beaches and salt marshes as main habitats (e.g., *Franzen*, *2006*; *Jaskuła*, *2011*; *Serrano*, *2013*). In the eastern Mediterranean it is distributed continuously all along the Adriatic and Aegean coasts, Turkish Straits and the Black Sea coastline (*Cassola & Jaskuła*, *2004*; *Jaskuła, Pešić & Pavicević*, *2005*; *Franzen*, *2006*; *Jaskuła*, *2007a*; *Jaskuła*, *2007b*). However, pronounced genetic structure with two divergent operational taxonomic units (OTUs) implies prolonged spatial isolation in the evolutionary history of this species. The observed level of divergence indicates that this isolation initiated an allopatric speciation. Their present distribution i.e., sympatry in the Pontic region reveals secondary contact of the already divergent lineages in this area. The Bayesian time-calibrated reconstruction of phylogeny shows that split between these OTUs begun in early Pleistocene. This coincides with beginning of recurrent glaciations resulting in eustatic sea level changes and climate aridisation that ever since dominated the global climate and landscape/habitat distribution (*Fagan*, *2009*). In the Mediterranean and in the Pontic region such global effects overlaid and strengthen the local effects of tectonic plate collision leading to Alpine orogeny, i.e., local land uplift and subsidence resulting in isostatic sea level changes, salinity fluctuations from freshwater to fully marine and habitat mosaicism (*Stanley & Blanpied*, *1980*). For example, during that time the connections of Pontic basin to Mediterranean Sea was lost and regained for more than a dozen times (*Kerey et al.*, *2004*; *Badertscher et al.*, *2011*). A profound impact of these events on the evolution and, hence, distribution of local both aquatic (*Audzijonyte, Daneliya & Vainola*, *2006*; *Nahavandi et al.*, *2013*) and terrestrial taxa

(e.g., *Böhme et al.*, *2007*; *Ferchaud et al.*, *2012*). We can assume that in case of *C. littoralis*, a halophilic species bound to coastal habitats, sea level fluctuations would significantly affect its distribution. The 2 Ma divergence time for *C. littoralis* OTUs derived from our data coincides with one particular disconnection of the Mediterranean and Pontic basins. At that time, from ca. 2 to ca. 1.5 Ma, the Meothic Sea, one of several predecessors of the Black Sea, turned into the predominantly freshwater Pontos Sea/Lake (*Grinevetsky et al.*, *2015*). This surely broke the formerly continuous stretch of coastal habitats connecting the two basins and thus, could be an effective barrier leading to split of *C. littoralis* population into the allopatric southern and northern lineages. Their detailed history is impossible to unravel, yet results of BSP analyses reconstructing past changes in effective population size indicate that both lineages started their demographic expansions at ca. 0.15 Ma. This coincides with the terminal stage of MIS-6, i.e., Wartanian/Saalian glaciation, and beginning of MIS-5e, i.e., Eemian interglacial (*Lisiecki & Raymo*, *2005*; *Marks*, *2011*). The latter was characterized by warmer climate and sea level higher by 6–9 m in comparison to Holocene (*Kopp et al.*, *2009*; *Dutton & Lambeck*, *2012*). In result, a wide connection between Mediterranean and the Pontic basin was re-established and the coastal habitats extended again, enabling exchange of faunas. Due to a deficiency of local studies, it is hard to compare our results to evolutionary history of any other terrestrial taxa in the area. However, a wealth of studies showing very similar spatiotemporal scenario in animal taxa comes from the coastal regions of the Gulf of Mexico and the adjacent Atlantic coast (summarised by *Avise*, *1992*). During Pleistocene, Cuba was connected with a land bridge to the Florida Peninsula what lead to divergence of populations of several terrestrial and aquatic animals, including also a local tiger beetle species *Cicindela dorsalis* Say, 1817 (*Vogler & DeSalle*, *1993*). Interestingly enough, however, according to our results both lineages are until now in the stage of demographic and spatial expansion, only the southern one has crossed the present Turkish straits. This asymmetry is hard to explain. Another interesting fact is that the isolation of Pontic basin from Mediterranean during the following Weichselian glaciation did not have probably any effect on the demography and phylogeography of the species. Based on the mitochondrial DNA marker only we cannot also conclude, whether the secondary contact of the divergent lineages effected in hybridization and or introgression. Answering this question requires employment of nuclear marker, what leaves a space for the future studies—much wider in terms of geographic coverage and molecular markers used.

Concluding, we have demonstrated that Pleistocene glaciations and associated sea level changes in the Mediterranean/Pontic region had a profound effect on the genetic diversity and distribution of widely distributed coastal insect species, generating some level of cryptic diversity. Our case study casts more light on the evolutionary relationships between populations of terrestrial animals inhabiting both the Mediterranean and Black Sea shorelines—a phenomenon that is still weakly explored in literature.

## ACKNOWLEDGEMENTS

The first author would like to thank to Iwona Jaroszewska, Piotr Jóźwiak, Błażej Pawicki, Maciej Podsiadło, Anna Stepień, and Bartosz Ukleja for their kind help in material collecting during the TB-Quest Expedition to the Balkans.

### Funding

The study was partly funded by the statutory fundings of the University of Lodz. The funders had no role in study design, data collection and analysis, decision to publish, or preparation of the manuscript.

### Grant Disclosures

The following grant information was disclosed by the authors:
The University of Lodz.

### Competing Interests

The authors declare there are no competing interests.

### Author Contributions

- Radomir Jaskuła and Tomasz Rewicz conceived and designed the experiments, performed the experiments, analyzed the data, contributed reagents/materials/analysis tools, wrote the paper, prepared figures and/or tables, reviewed drafts of the paper.
- Mateusz Płóciennik analyzed the data, contributed reagents/materials/analysis tools, wrote the paper, prepared figures and/or tables, reviewed drafts of the paper.
- Michał Grabowski conceived and designed the experiments, analyzed the data, contributed reagents/materials/analysis tools, wrote the paper, reviewed drafts of the paper.

### DNA Deposition

The following information was supplied regarding the deposition of DNA sequences:
The sequence data and trace files were uploaded to BOLD and subsequently also to GenBank (accession nos KU905171–KU905339).

### Data Availability

Data has been supplied in Table 1.

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
