# Peer review of "Pleistocene phylogeography and cryptic diversity of a tiger beetle, Calomera littoralis, in North-Eastern Mediterranean and Pontic regions inferred from mitochondrial COI gene sequences"

_PeerJ, doi:10.7717/peerj.2128_

## Round 0.1 · original submission · Minor Revisions

Thank you for submitting your manuscript to PEERJ.

I encourage you to read the reviews and the attached pdf file very carefully, and respond to each and every individual comment.

I do not agree with referee 2, regarding the use of a single marker of mitochondrial origin. The timeframe of the main questions addressed in the manuscript is compatible with the use of mitochondrial marker and does, therefore, not constitute a problem. Moreover, the authors clearly state that “Based on the mitochondrial DNA marker only we cannot also conclude, whether the secondary contact of the divergent lineages affected in hybridization and or introgression. Answering this question requires employment of nuclear marker, what leaves a space for the future studies – much wider in terms of geographic coverage and molecular markers used.” However, the geographical coverage of this work in comparison with the known geographical distribution of the species is a more serious problem, that needs to be clearly addressed.

If you are willing to send a thoroughly carefully revised version, I would be happy to reconsider your manuscript resubmission.

In your structured abstract, do not start the Discussion point with: “This might be due…”. State clearly what “this” is.

Reviewer 1 ·

Basic reporting

This manuscript is very nice contribution on the phylogeography of the tiger beetle species in the biodiversity important region. It is well structures, includes all the necessary components and meets the Scope and policies of the journal.
Figures are relevant, their resolution cannot be judged based on PDF, but I believe authors will provide suitable originals if the MS is accepted. Maybe the only exception is Figure 2. I know it is not the most important, but it would look much better, if it is redrawn with more patience. Now it looks like drawn with shaking hand, elytra are asymmetric etc. Regarding measurements shown in Figure 2, it is necessary to keep in mind that length is something from the beginning to the end and width from side to side, as marked in elytra. The same should be to head and pronotum, thus measurements 2 and 4 are lengths and 3 an 5 are widths. Also, I would suggest excluding mandibles from TBL, since they are movable, and so might cause inaccurate measures.
I was not able to look at the GenBank data, they were not available during preparing this review.
Finally, I must say I English is not my native language, however, I would strongly recommend authors to check it. It sounds to me as if I write it and I feel it is not very good.

Experimental design

Here I have no comments, the research is original, well described and interpreted.

Validity of the findings

In my opinion, findings described in this contribution are well supported, allowing authors to bring new evidence and new knowledge on the evolution of biota in the studied region.

Additional comments

Few comments were made in the PDF.

Annotated reviews are not available for download in order to protect the identity of reviewers who chose to remain anonymous.

Reviewer 2 ·

Basic reporting

see general comments below

Experimental design

see general comments below

Validity of the findings

see general comments below

Additional comments

Overall, this is an interesting study on Tiger Beetles in the eastern portion of the Balkans and Black Sea based on CO1 sequences.
This paper, however, suffers from sampling and methodological issues.
While sampling in the region is excellent, it only covers a small portion of the distribution of the species, letting the reader wonder what the entire story is.
The use of a single molecular mitochondrial marker is problematic. The use of CO1, a relatively conserved region, usually used for barcoding, not population genetics, adds to the less than ideal methodological approach. The lack of testing for evolutionary models and the use of Neighbor-Joining is common for barcoding studies, less so for population genetics studies.
Nevertheless, while results remain very interesting, they also raise some major questions. The presence of a southern haplotype in the north may be due to several scenarios (presence of sister species, hybridization, or a simple cline. This is a question that can easily be resolved using nuclear markers.
Two decades ago, such a study would have been truly fantastic, but it is currently below the standards of the field to leave this issue as an open question due to the use of a single molecular marker.
As much as I am not at all a champion of using multiple markers in all situations, the results presented here specifically require multiple markers to reach a definitive answer.

I am therefore very sorry not being able to recommend this study for publication at this stage.

---

## Round 0.2 · accepted · Accept

Thank you for your revisions of the manuscript in response to the referees suggestions. I have read through the manuscript and am satisfied with your response and revisions, such that I am happy to move the manuscript forward to production. I note that you did not change the starting sentence of the discussion section of the abstract, and I urge you to rephrase it.